# Grouting for Tunnel Stability Control and Inadequate Grouting Section Recognition: A Case Study of Countermeasure of Giant Karst Cave

**Peng Peng [1], Feng Peng [2], Zhenyu Sun [1] and Dingli Zhang [1,\*]**

[1] Key Laboratory of Urban Engineering of Ministry of Education, Beijing 100044, China
[2] Beijing Ruiwei Railway Engineering Co., Ltd., Beijing 100038, China
\* Correspondence: zhang-dingli@263.net

**Abstract:** Backfilling a giant karst cave with grouted engineering spoil as a new countermeasure for tunnels through giant karsts cave is studied in this paper. The numerical models of sections with different distribution characteristics of karst cave and tunnels are established for studying the deformation of surrounding rock and mechanical response of tunnel lining with and without grouting, respectively. The results illustrate that the countermeasure scheme is feasible. In order to ensure that the countermeasure can perform as expected effect, the inadequate grouting sections are recognized and verified using field grouting records and single-hole grouting quantity analysis. Finally, the application effect of the countermeasure scheme is evaluated by field monitoring of horizontal convergence. The result shows that the grouting can reduce the deformation of surrounding rock at the side wall and bottom of tunnel by 70–80% and reduce the stress redistribution range of surrounding rock. However, due to the great differences between the limestone and engineering spoil, the grouting cannot change the share of distribution of load between corresponding region surrounding rock, the max principle stress of tunnel lining is almost identical with and without grouting. The grouting reinforced engineering spoil backfill the giant karst cave can meet the requirement of excavation stability. The inadequate grouting sections caused by groundwater and through crack are identified effectively, and are verified by coring observation method. The horizontal convergence of the tunnel is less than 30 mm, and the stable state can be reached within 20 days, which demonstrate that the remarkable engineering results is achieved. The countermeasure of giant karst cave can provide a useful reference for similar project.

**Keywords:** engineering spoil backfilled; sidewall grouting; deformation control; grouting effect evaluation



## 1. Introduction

Karst landscape is a typical topographical features which is widely distributed in the world [1–3], according to the statistics, there are 15% land area of the world is covered with karst landscape [4,5]. The area of karst in China is approximately 3.44 million km$^2$, and is mainly dis-tributed in the Yunan province, Guizhou Province, Guangdong Province and Guangxi Province [1,6,7]. The karst cave is the result of long-term interaction between soluble rock and groundwater [8], tunnel-ing in the karst cave involves great challenge [9–12], such as water and mud inrush, tunnel infra-structure collapse and surface subsidence [12–14]. Sealing the through hydraulic crack and improving the carrying capacity by grouting is an effective and widely used method to ensure the safety of tunnel [15–17]. Therefore, scholars have carried out extensive and deeply researches on grouting for water plug and reinforcement.

For the Maanshan Park station project of Guangzhou metro Line 9, Cui et al. (2015) reduced the risk of water inrush and improved the safety of excavation by the comprehensive

methods containing backfilling the karst cave, building diaphragm wall and grouting sealing the hydraulic crack, and proposed the eight-step treatment method of karst tunnel [18]. Li et al. (2020) studied the mechanism and method of grouting sealing cracks through numerical simulation and experiment [19]. The results showed that the key to improve the sealing performance of karst tunnels is to reduce the groundwater flow rate. The influence degree of different factors on the sealing effect of grouting is obtained, the grouting quantity is considered to be the most important factor. Li et al. (2021) used numerical method to analyze the change law of groundwater seepage velocity and the pore pressure under the different groundwater flow rate [20]. The results indicated that the pore pressure and the retention rate of cement slurry increase with the decrease of seepage velocity. By incorporating the interval analytical hierarchy process into the technique for order preference by similarity to an ideal solution, a new risk identification model based on the fuzzy mathematics was proposed by Lin et al. (2021). The identification result was verified with the project of Maanshan Park station of Guangzhou metro Line 9. The proposed model can effectively identify the excavation risk and reduce accident occurrence [21]. By using two parameters of tunnel water inflow rate and quantized lining stability, Yau et al. (2020) proposed a probabilistic model of karst cavity distribution to evaluate the risk of tunnel excavation [22]. A numerical model was used for trial calculation, and good evaluation results are obtained. By using model test, Zhang et al. (2020) studied the proper-ties including the compressive strength, bending strength, gel time and viscosity of a new grouting material of potassium phosphate magnesium cement-based slurry, and the new grouting material is used to solve the problem of low grout retention rate in karst tunnel grouting [23].

However, the aforementioned research are based on the small karst cave, and the stability of rock strata between tunnel and small karst cave is the main topic. When the tunnel passes through a giant karst cave, the influence of engineering economy and size effect are quite significant [8]. The use of engineering spoil to backfill giant karst caves become a choice that taking count of the engineering economy and construction convenience. In this case, the governing factor of tunnel stability are transformed into the bearing capacity and sealing property of surrounding rock. Existing research [24,25] indicates that the engineering spoil is essentially a mixture which rock blocks is embedded in soil matrix, the mechanical property is extremely related to its microstructure and size [26–28]. Strongly inhomogeneity and discontinuity structure [29,30] of the engineering spoil lead to the existence of the through pores and poor physical mechanical properties [31,32]. When the engineering spoil is used as the surrounding rock of tunnel, the bearing capacity cannot afford the load thereby coordinate the mechanical response of the tunnel, and the water and mud inrush is easier to occur [33,34]. Therefore, in the engineering practice, when the tunnel passes through the giant karst cave backfilled with engineering spoil, it faces two problems. First, whether the backfill body can be grouted to improve its bearing capacity and sealing property, and second, how to determine the sufficient penetration of the slurry to ensure the effectiveness of grouting.

This paper investigates the geological and hydrological conditions of the Yujingshan tunnel of Chenggui Railway. The numerical simulation method is used to analyze the surrounding rock deformation and mechanical response of the primary lining before and after grouting. The feasibility and rationality of using grouting reinforcement to passes through the engineering spoil backfilled giant karst cave are theoretically explained. The grouting holes arrangement, grouting parameters and the grouting scheme adopted in the project are introduced. Based on detailed field grouting records, combined with single hole grouting quantity analysis and coring observation, the inadequate grouting sections are recognized, and the grouting reinforcement effect is verified by evaluated by field horizontal convergence monitoring.

## 2. Background of the Yujingshan Tunnel

The Yujingshan tunnel is a crucial component of the Chengdu-Guiyang high-speed railway in the Weixin county, Zhaotong city, Yunnan province, China. Figure 1 shows the

longitudinal section of Yujingshan tunnel and design profile of tunnel infrastructure. The tunnel is designed as 30‰ single-side uphill and one tube with two lanes tunnel, and the train speed limit is 250 km/h. As the Yujingshan tunnel is located in the transition region between Sichuan Basin and Yunnan-Guizhou Plateau, the terrain varies greatly in elevation, with a huge relative height difference of 50–650 m. The valleys of mountains are mostly southwest-northeast trending, which is consistent with the direction of the structure line. The tunnel is located at the downstream of Nanguang River, which is one of tributary of Yangtze River. The tunnel is mainly composed of east-west geological structures, crossing the structure line at different angles, the whole tunnel section passes through four faults [8]. Most sections of Yujingshan tunnel is buried at a depth of 200–300 m. The tunnel mainly passes through limestone, sandstone, mudstone, tuff and dolomite. For the section from mileage number of D3K279 + 865–D3K279 + 956, the tunnel passes through the giant karst cave hall. According to the geological survey report, the tunnel infrastructure with dimensions of 17.22 m × 25.21 m. The study region can be divided into two layers, the first layer is dominated by the fragment stone, and the second layer is composed of limestone.

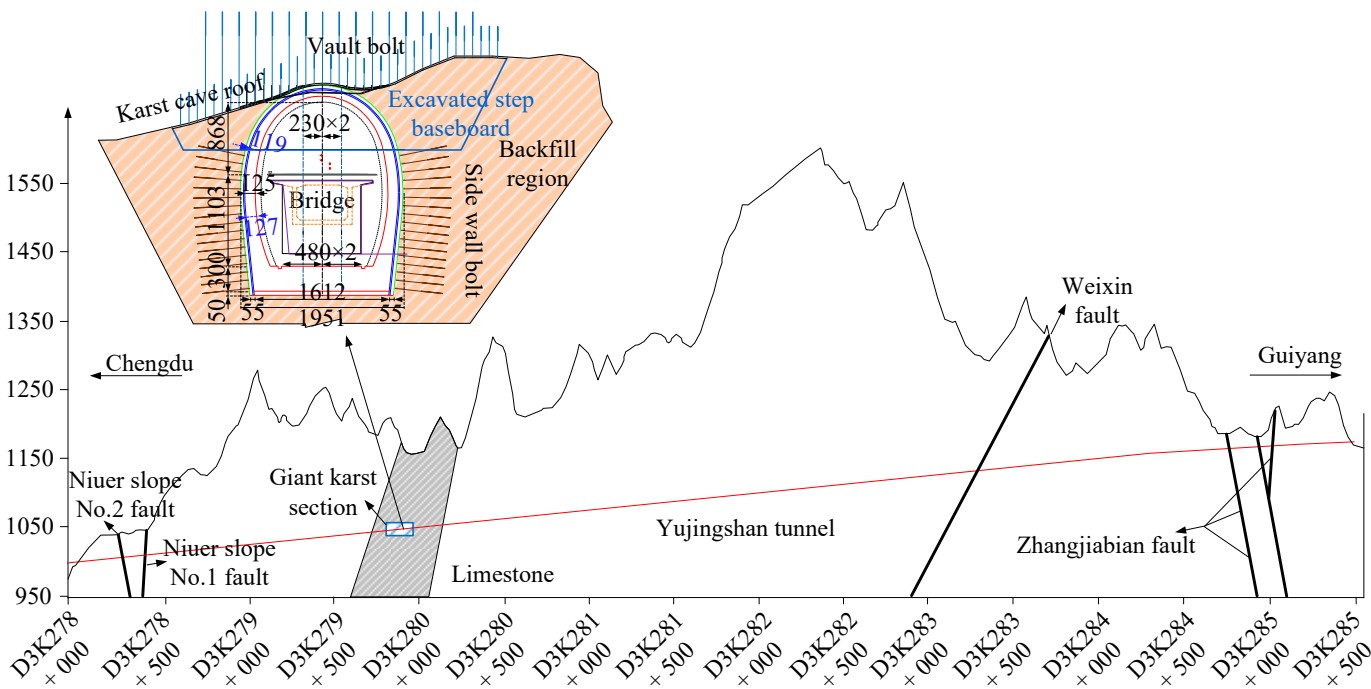

**Figure 1.** Design profile of tunnel infrastructure.

The length of the tunnel is 6306.28 m from the entrance mileage D3K277 + 860 to exit mileage D3K284 + 164, and the thickest overburden is 350 m. The auxiliary excavation of parallel pilot tunnel is adopted, and construction organization is divided into three parts according to the entrance work region, parallel pilot tunnel work region and exit work region in Yujingshant tunnel. The entrance region and exit region employed open excavation method, and the New Austrian Tunneling Method (NATM) is adopted in the underground excavation section of the parallel pilot tunnel region. The geological survey shows that the Yujingshan tunnel mainly cross the coal measure strata and soluble rock strata, where the mileage number from D3K278 + 990 to D3K279 + 061 contains high gas concentrations, the mileage number from D3K279 + 061 to D3K279 + 500 has coal and gas outburst layers, and the Lower Permian Qixia Maokou formation is founded in the range of mileage number from D3K279 + 500 to D3K280 + 290. In July 2016, when the parallel pilot tunnel region was excavated along the small mileage direction of the tunnel to D3K279 + 872, an unprecedented huge karst cave in the history of tunnel construction of China was found. Geological survey shows that the karst cave hall is almost 100 m along the lane, and the width is almost 230 m. The roof of karst cave is present as the dome shape,

and the bottom of the karst cave developed on the Yujingshan underground river. Due to the extremely complicated construction condition of the karst cave, the excavation risk is intensely high.

Figure 2 shows the geological profile of karst cave, the location of the Yujingshan tunnel developed Niuer slope No. 1 fault, Niuer slope No. 2 fault and Weixin fault, and the Yujingshan tunnel is mainly astride the roof of the karst cave. The geology condition of the karst cave region is dominated by soluble limestone, and the bottom of the karst cave hall filled with talus body which shows strongly layered characteristic. The height of the talus body is almost 78 m and 100 m on the left and right side, respectively, and the width of the talus body is nearly 134 m. According to the engineering construction scheme, the karst cave is backfilled with engineering spoil.

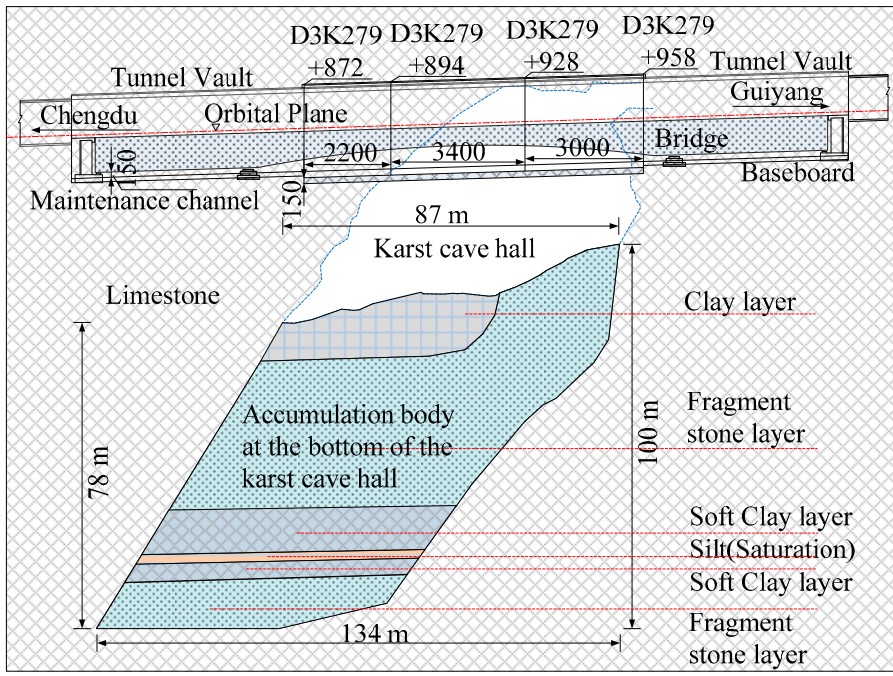

**Figure 2.** Geological profile of karst cave.

The hydrological conditions around the tunnel are shown in Figure 3. The location of the Yujingshan tunnel is mainly distributed on Chishui-river, Nanguang-river and Zhaxi reservoir. Field survey shows that the hydrogeological environment of Yujingshan tunnel is mainly consist of surface water and underground water. The surface water is seasonal stream water, which is recharged by atmospheric precipitation and greatly affected by season. The underground water is bedrock fissure water and karst water. In the limestone layer, Yujingshan underground-river is developed, whose length is almost 18 km, and the angle of intersection with the direction of large mileage is 58°. According to the field survey report, there are 5 proved inlet total flux is 1.2–5.0 m³/s, and 9 outlets, the maximum observed flux is about 21.16 m³/s [35,36].

Due to the giant size of the karst cave and the extremely complicated geological and hydrological environment of the tunnel, the comprehensive countermeasures containing underground river diversions, karst cave backfill, concealed excavation and bridge crossing method are adopted in the Yunjingshan tunnel giant karst cave. In the karst cave backfill construction stage, the giant karst cave is backfilled in the order of construction from low position to high position, and from side wall of the cave to the middle. The backfilled surface is compacted by mechanical leveling every 3 m to ensure that the backfilled strata provide bearing capacity as much as possible. In order to effectively discharge the groundwater seeping out from the cavern wall, the karst cave wall is backfilled with blocks with a diameter of not less than 0.3 m within 3 m from the karst cave wall, and

the rest of karst cave is backfilled with engineering spoil. During underground tunneling construction, the sidewall penetration grouting is used to reinforce the surrounding rock of Yujingshan tunnel.

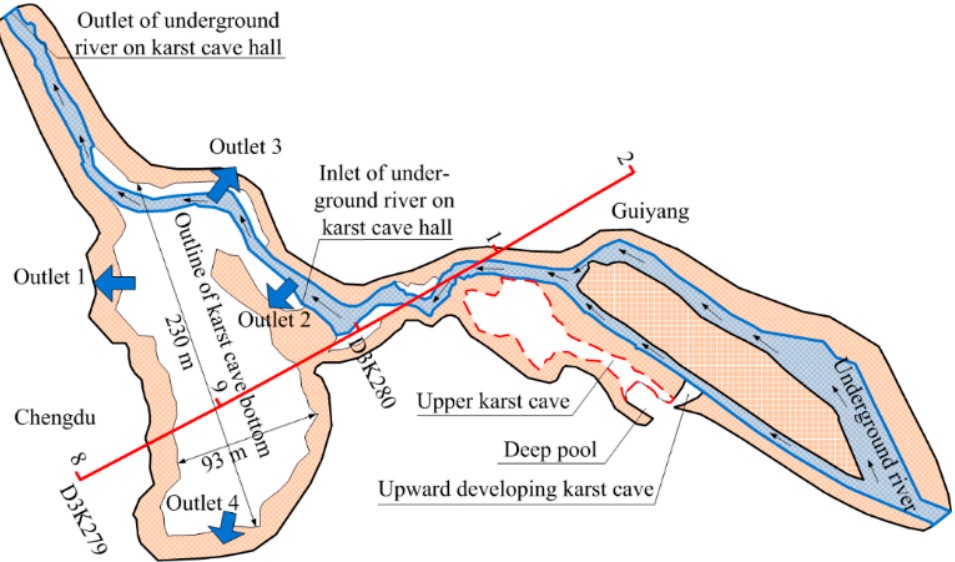

**Figure 3.** Hydrological profile of karst cave hall.

## 3. Numerical Simulation of Tunnel Excavation without Grouting

### 3.1. The Selection of the Tunnel Section and Establishment of Numerical Model

In order to prove the feasibility of tunnel construction in grouted reinforced engineering spoil, the numerical simulation of Yujingshan tunnel is conducted in FLAC$^{3D}$. Due to the complicated geology conditions of Yujingshan tunnel, the mechanical response of a single tunnel section cannot truly reflect the mechanical response of tunnel infrastructure. Therefore, the tunnel sections are generally divided into two types according to the positional relationship between the karst cave and tunnel infrastructure. A typical section of the first kind is similar to the section of mileage number D3K279 + 888 (I-type); the karst cave is on the single side of the tunnel infrastructure, which leads to large asymmetrical load on the tunnel lining. The other kind is similar to D3K279 + 938 (II-type). Although the tunnel infrastructure is surrounded by the karst cave except for the crown of the tunnel, the large width of the karst cave may cause large deformations of the tunnel basement. Figure 4 shows the numerical model of the tunnel with two sections. The model dimensions of mileage number D3k279 + 888 and D3k279 + 938 are 120 m × 50 m × 140 m and 140 m × 50 m × 140 m, respectively. The dimensions of tunnel infrastructure are 17.22 m × 25.21 m. The mechanical response of surrounding rock under a three-step excavation method is analyzed by Xie et al. [37], the results show that the small stress is induced in the second lining, the stability of surrounding rock can meet the requirement of safety. Huang et al. (2022) proposed a countermeasure against the complicated tunnel with inclined shafts, connecting air ducts and main tunnel excavation in limestone region with low bearing capacity [38]. The settlement of complex tunnel vault is controlled by flexibly converting the excavation method. Compared with the literature, the three-step excavation method was suitable for tunneling in a limestone region and was adopted during the simulated analysis. According to the geological survey report of Yujingshan tunnel, the physical and mechanical parameters of strata in the giant karst cave region are listed in Table 1. The mechanics properties of the V-level surrounding rock are adopted to the engineering spoil. The Mohr-Coulomb failure criterion is adopted in the surrounding rock, the second lining is considered to the elastic model.

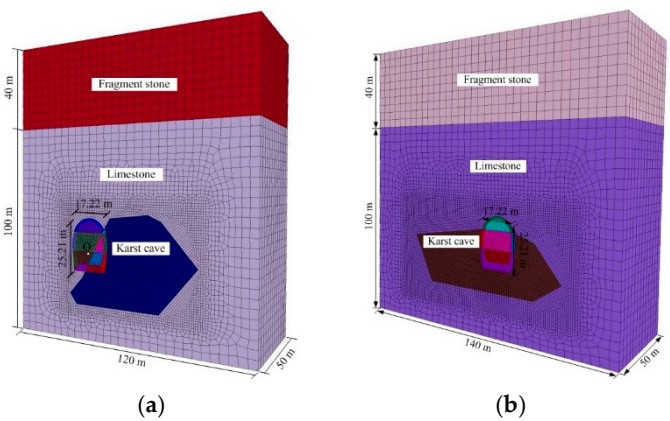

**Figure 4.** The numerical model of tunnel with (**a**) D3k279 + 888 section and (**b**) D3k279 + 938 section.

**Table 1.** The physical and mechanics parameters of stratum.

| Name | Bulk Modulus /MPa | Shear Modulus /MPa | Cohesion /kPa | Friction Angle /° | Density /kg/m³ |
|---|---|---|---|---|---|
| Fragment stone | 43.01 | 16.49 | 0 | 30 | 2300 |
| Limestone | 4340 | 2360 | 800 | 60 | 2650 |
| Engineering spoil (compaction state) | 21.52 | 10.63 | 20 | 12 | 2140 |
| Shotcrete | 29,500 | 12,300 | - | - | 2400 |
| Second lining | 35,000 | 13,500 | −900 s | - | 2400 |
| Grouting reinforced engineering spoil | 150 | 75 | 110 | 38- | 2800 |

The grouted bolt, anchor mesh shotcrete and steel arch compound support are adopted to the first lining, and the mechanical properties of support are listed in Table 2. The primary lining and bolt are simulated by shell and cable structure element, respectively. The steel mesh and steel arch convert to the strength of the primary lining according to the equivalent strength principle, that is,

$$E_c = E_0 + \frac{A_s E_s}{A_c} \tag{1}$$

where $E_c$ and $E_0$ are the converting elasticity modulus and origin elasticity modulus of concrete, respectively. $E_s$ is the elasticity modulus of steel, and $A_s$ and $A_c$ are the sectional area of steel arch and concrete, respectively. The elasticity modulus and Poisson ratio of bolt are 210 Gpa and 0.3, respectively.

**Table 2.** Parameters of support.

| C30 Shotcrete Thickness /cm | Grouted Bolt (arch/3.0 m) | | Grouted Bolt (Side Wall /5.0 m) | | Steel Mesh | | Steel Arch | |
|---|---|---|---|---|---|---|---|---|
| | Model | Space /(m × m) | Model | Space /(m × m) | Model | Space /(m × m) | Model | Space /m |
| 20 | Φ32 | 1.2 × 1.0 | Φ32 | 1.2 × 1.0 | Φ6 | 0.2 × 0.2 | I20b | 1.0 |

When the tunnel is excavated to the karst region, the karst cave is backfilled with engineering spoil. In this case, the karst cave is not disturbed by external forces and remains stable. Therefore, the karst cave wall should not be directly connected with engineering spoil. In this paper, the interface elements between the karst cave wall and engineering spoil are established. The mechanical properties of interface elements can be calculated by

referring to the physical properties of the strata around the interface and combining the following formula:

$$k_n = k_s = 10\max\left(\frac{K_t + \frac{4}{3}G_t}{\Delta z_{\min}}, \frac{K_b + \frac{4}{3}G_b}{\Delta z_{\min}}\right) \tag{2}$$

where $K_t$, $K_b$, $G_t$ and $G_b$ are the bulk and shear modulus of the strata located at the upper and lower sides of the interface, respectively. $\Delta z_{\min}$ is the minimum size of the contact region along the direction of the normal. $k_n$ and $k_s$ are the equivalent coupled spring stiffness of normal and tangential direction, respectively.

### 3.2. The Validation of Numerical Model

In order to verify the accuracy of the numerical model, the horizontal convergence of I-type and II-type section with grouting are simulated studied and compared with the field monitor results. Figure 5 shows the simulated and field monitor results of horizontal convergence, respectively. It is illustrated that the horizontal convergence of upper, middle and lower step shows the variation character of 'rapid-gentle' increasing and then reaches a stable state. The error values between the simulated and field monitor results for the upper, middle and lower steps are 31.4%, 9.5% and 20.62%, respectively. Because the top of the tunnel infrastructure is in direct contact with the giant karst wall, the karst cave is affected by groundwater. Therefore, the surrounding rock on the top of the tunnel is softened by groundwater, resulting in a large error between the simulated and monitoring results.

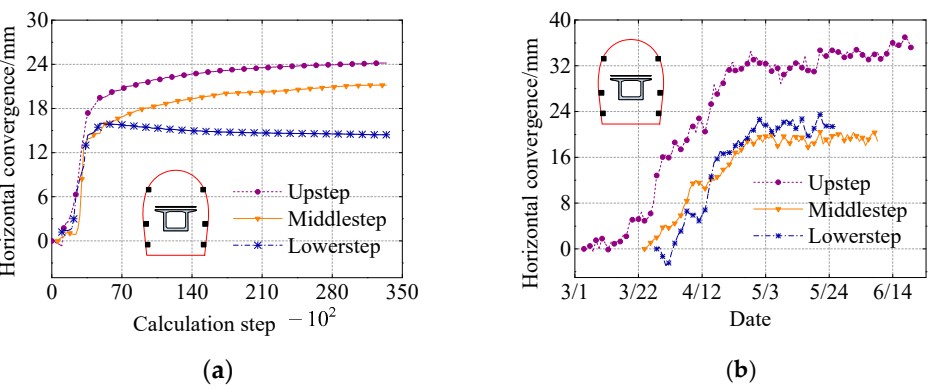

(**a**)　　　　　　　　　　　　　　　　　　　　　　(**b**)

**Figure 5.** Horizontal convergence (**a**) simulated and (**b**) field monitor results of I-type tunnel.

Figure 6 shows the simulated and field monitor results of horizontal convergence of II-type section tunnel, respectively. For the simulated results, the upper and lower steps show the tendency of increasing rapidly first and then slowly. However, the field monitor results show that the horizontal convergence remains stable after rapid growth, the error values of up and lower step between simulated and field monitor results are 19.26% and 7.73%, respectively. The horizontal convergence of middle step is first rapidly increased and then slowly, with an error of only 6.35%. The results show that the mechanical response of Yujingshan tunnel can be effectively simulated by established numerical model.

### 3.3. Mechanical Response Analysis of I-Type Section Tunnel

Figure 7a shows the simulated results of surrounding rock deformation on Y = 10 m section with and without grouting in I-type tunnel, where the arrow points to the deformation direction. Without grouting, large deformations are occurred in A and B region in the backfill body. The maximum deformation is 82.56 mm at point C in the A region, which is caused by the poor physical properties of the engineering spoil. The maximum deformation at D in B region is 55.25 mm. The reason is that the B region is connected to the karst cave wall, after the large displacement occurs in A region, the cumulative effect of displacement and the poor mechanical properties of the engineering spoil cause the backfill body produces the landslide phenomenon. Figure 7b shows the deformation of surrounding rock of I-type section with grouting, the displacement of the surrounding rock

is small and uniform. Although the cumulative effect of displacement causes the landslide in the E region, the maximum displacement is only 31.27 mm. The settlement of tunnel surrounding rock, consist of uniformity and isotropy limestone, is analyzed by Ding et al. and Xu et al. [39,40]. The results show that large vertical displacement occurs at the tunnel vault, and small displacement is induced at the invert arch of the tunnel. However, due to the existence of the weak region of engineering spoil backfilled giant karst cave, the maximum displacement is occurred at the contact region of limestone region and backfilled region, which is consistent with the literature result made by Zhang et al. [41].

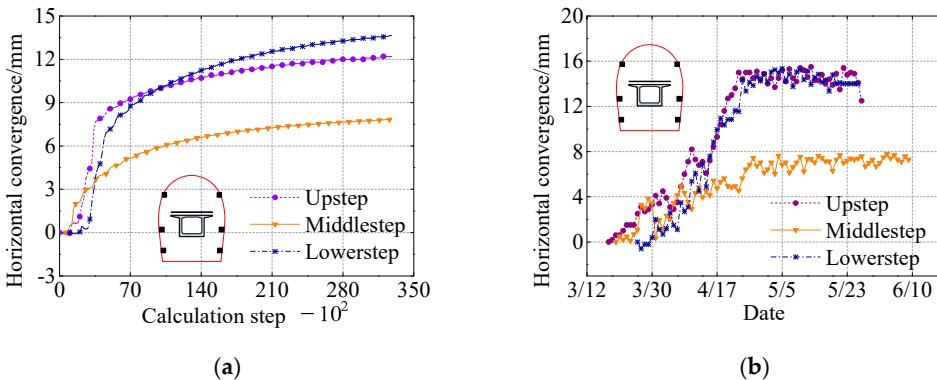

(**a**)  (**b**)

**Figure 6.** Horizontal convergence of II-type tunnel (**a**) numerical simulation result (**b**) Field monitor result.

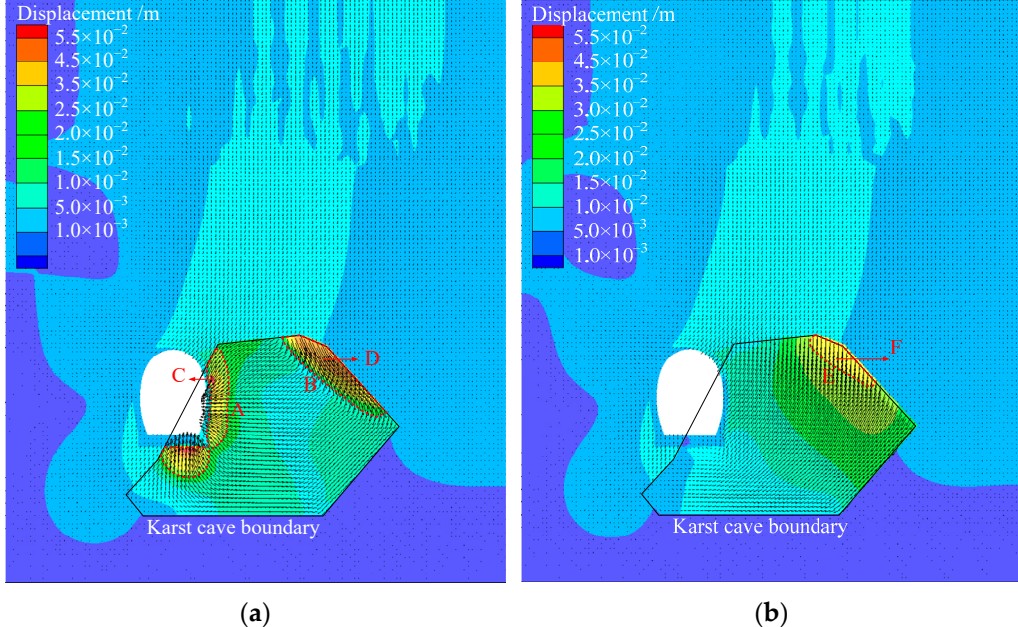

(**a**)  (**b**)

**Figure 7.** Displacement of surrounding rock of I-type tunnel (Y = 10 m section) (**a**) without grouting (**b**) with grouting.

Figure 8a shows the bending moment distribution on the Y = 10 m section before and after grouting in the I-type tunnel, where K and M are the intersections of the karst cave border and the tunnel primary support. The primary support is divided into three regions according to the karst cave boundary. The upper left corner of the karst border is the limestone region, the lower right corner of the karst cave border is the backfilled region, and the contact region is close to K and M. Before grouting, the primary support in the backfill region produces a large bending moment. The maximum moment with a value of 144.03 kN × m occurs at the right side of the tunnel bottom, while the primary support in the limestone region produces a small bending moment, and the maximum moment is

located at the upper right arch shoulder, only 54.43 kN × m. Large bending moment is generated at the primary support in the contact region, and the maximum positive and minimum negative bending moments around K are 88.98 kN × m and −26.20 kN × m, respectively, and the maximum positive and minimum negative bending moments around M are 68.44 kN × m and −23.22 kN × m, respectively, indicating that large shear force is produced in the contact region. After grouting, the bending moment of the primary support is improved obviously, and a large bending moment of −34.17 kN × m is generated only at the right arch foot, while the primary support within the limestone remains almost unchanged. The maximum positive and minimum negative bending moments occurred around K are 132.29 kN × m and −51.67 kN × m, respectively, and the maximum positive and minimum negative bending moments occurred around M are 80.13 kN × m and −50.53 kN × m, respectively. The increase of bending moment in the contact region after grouting indicates that a greater shear force is generated in the contact region. This is due to the enhanced mechanical properties of the backfill body, resulting in a greater load than the surrounding rock in the backfill body, while the shear stress in the limestone region remains unchanged, the greater shear force is produced.

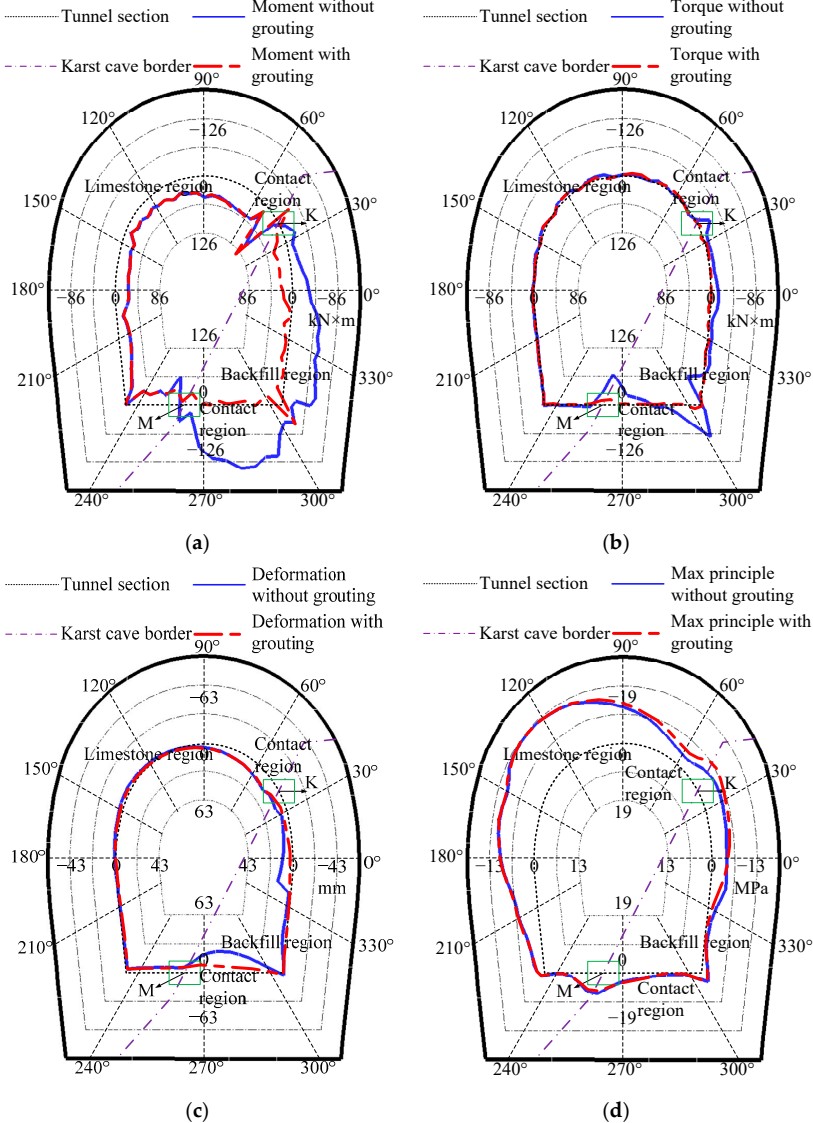

**Figure 8.** Mechanical response of I-type tunnel on Y = 10 m section (**a**) bending moment (**b**) torque (**c**) deformation and (**d**) max principle of support.

Figure 8b shows the torque distribution of Y = 10 m section before and after grouting in the I-type tunnel. The torque in the limestone region is almost 0, while large torque is generated in the backfill and contact region, where the minimum negative torque of $-79.95$ kN $\times$ m is occurred at the right arch foot, and the maximum positive torque is produced at the M with a value of 66.71 kN $\times$ m. After grouting, the torque distribution at the primary support is almost 0. Calculate the torque distribution by using Kirchhoff thin plate theory

$$M_{yx} = \int_{-\frac{t}{2}}^{\frac{t}{2}} z\tau_{yx}dz = -\frac{Et^3}{12(1-\mu^2)}\frac{\partial^2 w}{\partial x \partial y} \tag{3}$$

where $E$ and $t$ are the elastic modulus and thickness of primary support, $y$ is the direction along the tunnel excavation, $z$ is the direction of the plate thickness, $x$ is the direction determined by the right-hand rule, $\mu$ is the Poisson's ratio, and $w$ is the deflection of the plate. Equation (3) shows that the torque of the primary support is related to the deflection. In the current tunnel support design method, the effect of torque is often neglected, because the tunnel is considered as a plane stain problem in the design. Therefore, $\partial w/\partial x = 0$ in the x-direction and the torque is 0, the effect of torque can be ignored in the design process. When the tunnel is excavated in the karst cave backfilled with engineering spoil, the mechanical property of the backfill body is very weak, resulting in complex forces on the primary structure and generating a non-negligible torque. After grouting, the deformation along the x-direction decreases and the torque decreases to negligible, and the effect of torque on the primary support can be disregarded.

Figure 8c shows the deformation of the primary support on the Y = 10 m section before and after grouting in the I-type tunnel. The deformation of the primary support in the limestone region is almost 0, indicating that the limestone has good mechanical properties. The surrounding rock in the limestone region can withstand large loads during the stress redistribution caused by the tunnel excavation, which improves the mechanical characteristics of the primary structure. In the backfill region, the maximum deformation generated in the tunnel bottom and second step foot are 23.61 mm and 18.64 mm, respectively. The maximum deformation generated in the first step foot K is 17.25 mm. The reason for the large deformation in the tunnel bottom is that the bearing capacity is insufficient due to the weak mechanical properties of the surrounding rock, and the primary support bears a large load. The large deformation at the excavation step foot is the result of the stress release caused by the lagging construction of the primary support. After grouting, the improvement of the mechanical properties of the surrounding rock improves the bearing capacity, which reduces the primary support deformation at the tunnel bottom to 0. The large deformation of the primary support at the excavation step foot is decrease, the maximum deformation are only 12.35 mm at K.

Figure 8d shows the max principle distribution on the Y = 10 m section in the I-type tunnel, the negative value means to the compression state of the primary support. The large maximum principle is distributed in the limestone region, the maximum value occurs at the left tunnel shoulder of $-17.40$ MPa. The maximum principle in the backfill region is small, the maximum value in the backfill region are only $-5.77$ MPa. The reason is the release of stress and lagging construction of the primary support resulting in a large load on the surrounding rock in the limestone region and small load in the backfill region. When the primary support is constructed, a large maximum principle stress is generated on the support structure in the limestone region. In the contact region, the maximum principle in K and M are $-5.55$ MPa and $-7.81$ MPa, the large maximum principle stress generated in the contact region are the result of large shear forces due to the stiffness mismatch. After grouting, the maximum principle stress distribution on the primary support in the limestone region and backfill region remains unchanged, which is due to the fact that the enhancement of the mechanical properties of the backfill body by grouting is not sufficient to change the law of load distribution. In the contact region, the maximum principle stresses at K and M increases to $-7.89$ MPa and $-14.80$ MPa, respective, due to the increasing of shear force.

*3.4. Mechanical Response Analysis of II-Type Section Tunnel*

Figure 9 shows the displacement simulated result of II-type tunnel on the Y = 10 m section before and after grouting, where the arrow pointing to the direction of displacement. Large displacement is produced in three regions A', B' and C' in the backfill body, where the maximum displacement in A' region occurs at E', which is 414.52 mm, and the maximum displacement in B' region is located at D', with a value of 581.09 mm. The reason for the large displacement on the side wall of tunnel is the poor mechanical properties of the engineering spoil. Large displacement is produced under the small stress, and the lagging construction of primary support cannot restrain the surrounding rock displacement in time. The maximum displacement of the C' region at the tunnel bottom occurring at F', with a value of 66.06 mm. The reason is that after excavation, with the stress release of the surrounding rock, the surrounding rock at the tunnel bottom is squeezed by the surrounding rock on both sides of tunnel, resulting in upward development. After grouting, a large displacement occurred in the G' region, which is caused by the contact connection between the backfill body and karst cave wall, and a beam effect occurred in the karst cave wall. As a result, the span has a large deflection, and the backfill region has a large displacement due to the contact stress. However, the displacement of surrounding rock is obviously decreased, the maximum displacement in the right and left of the tunnel are located at H' and J', which are 29.48 mm and 21.67 mm, respectively.

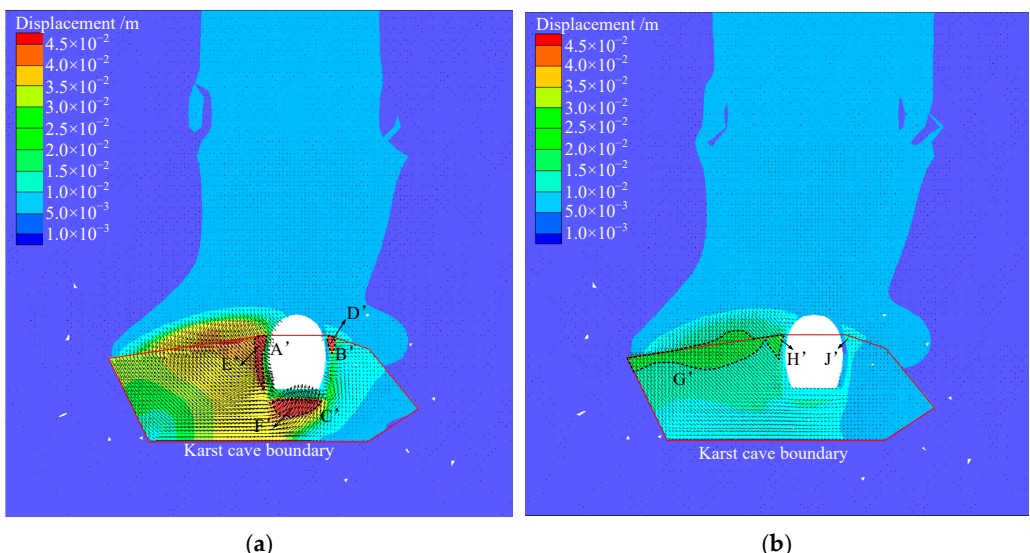

**Figure 9.** Displacement of surrounding rock of II-type tunnel (Y = 10 m section) (**a**) with grouting (**b**) without grouting.

Figure 10 shows the mechanical response of II-type tunnel on Y = 10 m section. K' and M' are the intersection between the primary support and the karst cave border on left and right side of the tunnel, respectively. According to the karst cave boundary, the primary support is divided into three regions, one is the limestone region above the boundary, the second is the backfill region below the boundary, and the third is the contact region between the limestone and the backfill region around the K' and M'. As shown in Figure 10a, in the limestone region, the bending moment gradually increases from the tunnel vault to the K' and M', and the maximum bending moment on the left and right sides of the tunnel are 32.03 kN × m and 23.32 kN × m, respectively. Large bending moment are generated in the contact region, where the maximum positive and minimum negative bending moments around K' are 63.37 kN × m and −21.55 kN × m, respectively, the maximum positive and minimum negative bending moments around M' are 32.31 kN × m and −24.18 kN × m, respectively. These are indicated that large shear forces are generated around the K' and M', resulting in bending moment changes. In the backfill region, large bending moments are produced at the tunnel arch foot and the tunnel bottom, and the bending

moments at the left and right tunnel arch foot are −78.34 kN × m and −50.97 kN × m, respectively, the maximum bending moment at the tunnel bottom is −99.14 kN × m. After grouting, in the contact region, the maximum positive and minimum negative bending moments around K' are increasing to 65.26 kN × m and −26.25 kN × m, respectively, and the maximum positive and minimum negative bending moments around M' are increasing to 40.58 kN × m and −24.35 kN × m. These are indicated that the larger shear force is generated in the contact region, because the grouting improves the bearing capacity of surrounding rock in the backfill region, and the shear force increases, but the shear force in the limestone region remains constant. Due to the stress distribution characteristics of the tunnel primary support, the large bending moments occurred at the left and right tunnel arch foot are −27.29 kN × m and −22.43 kN × m.

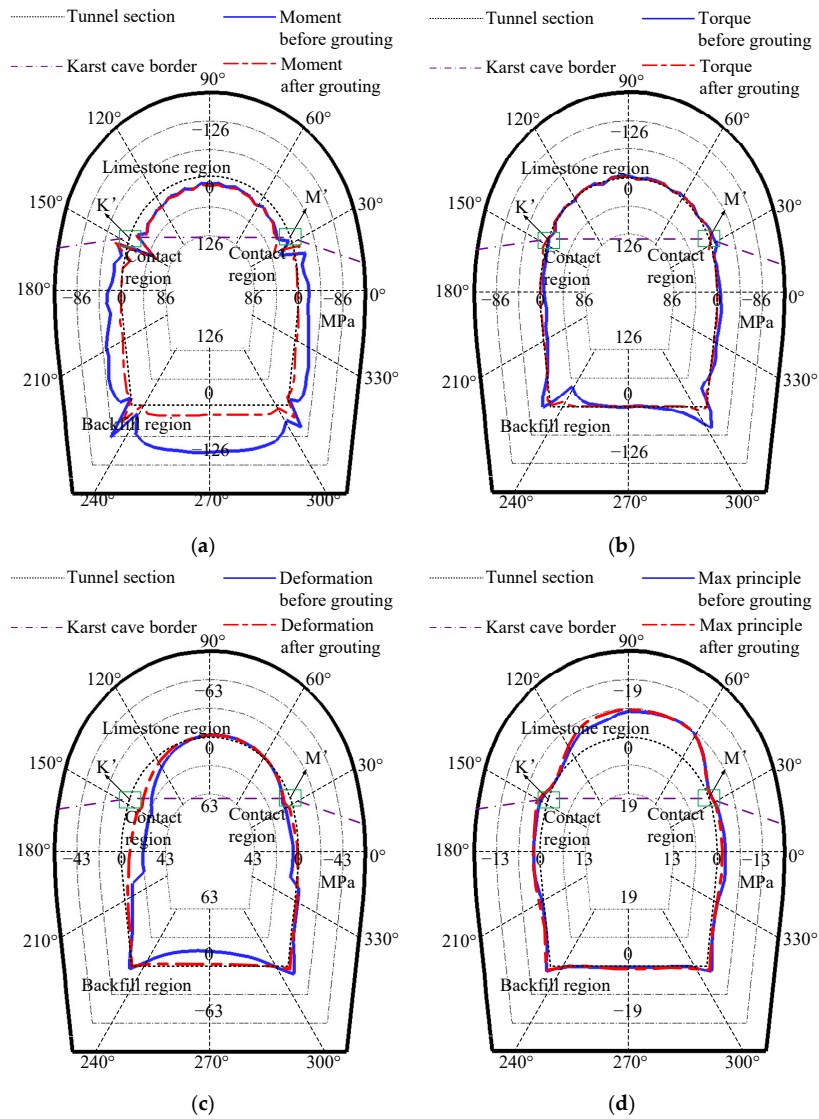

**Figure 10.** Mechanical response of II-type tunnel on Y = 10 m section (**a**) bending moment (**b**) torque (**c**) deformation and (**d**) max principle of support.

Figure 10b shows the torque distribution before and after grouting on the Y = 10 m section in the II-type. Before grouting, the large torque distributed at the left and right tunnel arch foot are 50.46 kN × m and −55.32 kN × m, respectively. Due to the symmetry of the strata on the left and right side of tunnel, there is no obvious torque at other location on the tunnel section. However, at the tunnel arch foot, due to the structural characteristics, a large load and complex stress state are occurred, resulting in large torque distribution.

After grouting, the torque on the tunnel section is 0, indicating that grouting can effectively improve the torque distribution.

Figure 10c shows the deformation of the primary support before and after grouting in the Y = 10 m section of II-type tunnel. K' and N' are the first excavation step and second excavation step boundary, respectively. The tunnel vault is in the limestone region, and the deformation of tunnel vault is almost 0 due to the good mechanical property of the surrounding rock. A large deformation occurred at the left step feet of the first and second excavation step, in which the displacement at K' and M' are 30.22 mm and 36.04 mm. The reason is that there is a large area of the karst cave on the left of tunnel. After excavation and before primary support construction, the stress of surrounding rock is fully released, resulting in a large deformation at the feet of the excavation step after the construction of primary support. After grouting, the deformations on the K' and N' are 16.95 mm and 11.78 mm, respectively, the deformation of primary support is effectively improved by grouting.

Figure 10 shows the maximum principle distribution before and after grouting on Y = 10 m section of II-type tunnel. Before grouting, a large maximum principle is generated in limestone region, while the maximum principle stress in the backfill region and contact region is small, where the maximum principle stress in the contact region is almost 0. After grouting, the distribution law of the maximum principle stress is almost constant, which is the same as that of I-type tunnel.

## 4. Grouting Design and Scheme

### 4.1. Grouting Penetration Distance

The basic premise of an economical and effective grouting scheme is the reasonable value range of grout penetration distance. Although scholars proposed many theoretical equations based on the different considering of influence factors such as filtration effect and tortuous length etc., the columnar penetrate equation is recognized by most researchers, and its expression is

$$R = \sqrt{\frac{2kpt}{n\beta \ln \frac{R}{r_0}}} \tag{4}$$

where $R$ is the penetrate radius (centimeter), $r_0$ is the radius of grouting hole (centimeter), $p$ is the pumping pressure (centimeter head), $k$ is the permeability (cm/s), $t$ is the grouting time (s), $n$ is the porosity, and $\beta$ is the ratio of slurry viscosity to water viscosity. In this project, $r_0$ can be taken as 5.4 cm, other parameters are: $k = 4.8 \times 10^{-4}$ cm/s, $n = 0.10$, $\beta = 2.50$, grouting time $t$ are 900 s. The pumping pressures are assumed as $2.0 \times 10^4$ cm head and $3.0 \times 10^4$ cm head, the penetrate distances are 1.61 m and 1.91 m, respectively. The field grouting tests are conducted to obtain a reasonable grouting penetration distance. The results shows that when the concentration of suspension slurry is 0.5, grouting time is 900 s and the grouting pressure is 2.0 MPa. For the 50 mm and 108 mm grouting holes, the longest grouting penetration distances are 1.85 m and 2.15 m, respectively. By comparing the penetration distance results of different methods, 1.5–2.1 m is finally considered as the design of grouting hole arrangement in order to make the grouting fully diffuse in the stratum and bite each other. Assuming that the soil skeleton cannot be compressed, the void space in the grouted region is filled with slurry, the total slurry volume can be expressed as

$$Q = \pi R^2 \cdot n \tag{5}$$

where $R$ is the radius of grouted region. The grouted region is assumed as a columnar and the grouting volumes are 1.02 m³/m and 1.26 m³/m. In the actual construction process, the grouted region shows different properties due to the different position of injected media, the grouting time and pumping pressure should be adjusted according to the actual engineering condition.

### 4.2. Yujinghsan Tunnel Side Wall Grouting Reinforcement Scheme

Due to the limitation of construction conditions, the back step grouting is used for construct penetration grouting in the backfill region. The grouting parameters listed in Table 3 are adopted in grout engineering of Yujingshan tunnel side wall. In consideration of environment protection and the safety of drinking water for surrounding residents, the materials used in this project are mainly sulphate aluminium cement (SAC) and cement-sodium silicate (C-S).

**Table 3.** Penetration grouting parameters.

| Index | Value | Index | Value |
|---|---|---|---|
| Stop pressure | 2–3 MPa | Backfill material ratio | Water: Cement: Soil = 1.6:1:1 |
| Penetration distance | 1.5–2.1 m | Backfilled rubble | 5–10 mm |
| Grouting flow | 77–165 L/min | Back step distance | 1 m |
| Single slurry ratio (SAC) | W:C = (0.6–1):1 | Cement-sodium silicate slurry (C-S) | W:C:S = (0.6–1):1:1 |

The arrangement of grouting holes shown in Figure 11 are based on the results of theory calculation and in-situ grouting test. The arrangement of grouting holes depends on the positional relationship between the karst cave and the tunnel infrastructure. 108 mm and 50 mm sleeve valve pipes are used for grouting in the left and right region of tunnel axis, respectively. The grouting holes around the reinforcement region are mainly grouted with cement-sodium silicate slurry to form the stop slurry curtain. In the central reinforcement region, the SAC slurry is used to improve the structural strength. In addition, the curtain for slurry stopping is adopted in D3K279 + 872 section right lane, D3K279 + 913 section and D3K279 + 955.5 section left lane. Due to the weakness mechanical properties of engineering spoil and crush of surrounding rock, grouting holes collapse frequently during grouting hole drilling, which seriously affects the construction progress. The casing drilling method is used to improve the construction efficiency of the grouting holes.

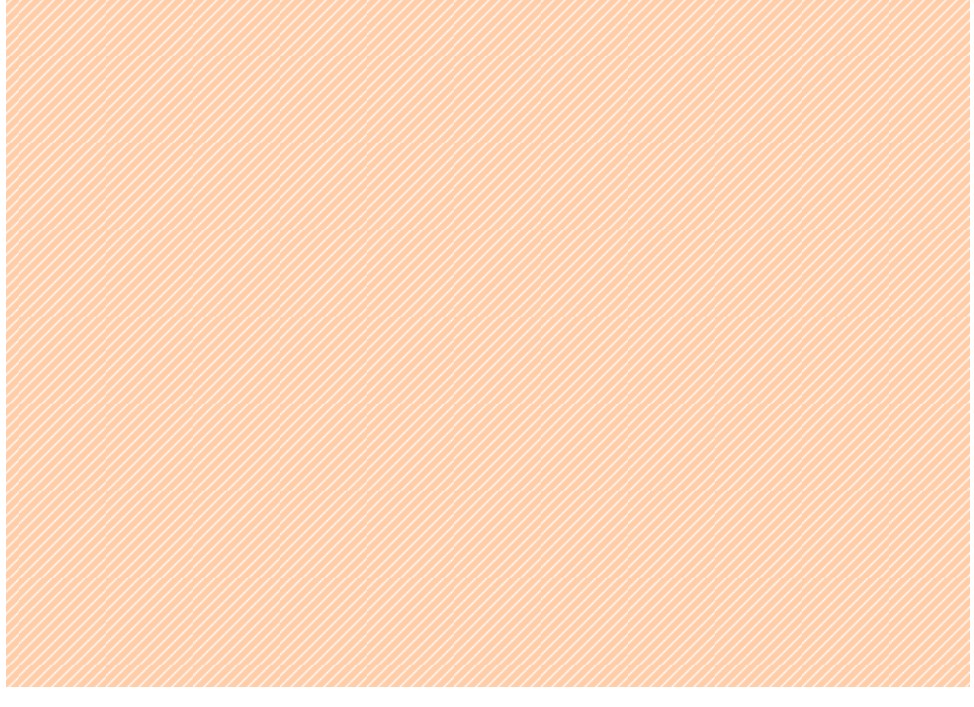

**Figure 11.** The arrangement of grouting holes.

Due to the irregular nature of the karst cave section, the grouting drilling design is various in different tunnel sections. Figure 12 shows the grouting drilling design in typical sections of Yujingshan tunnel.

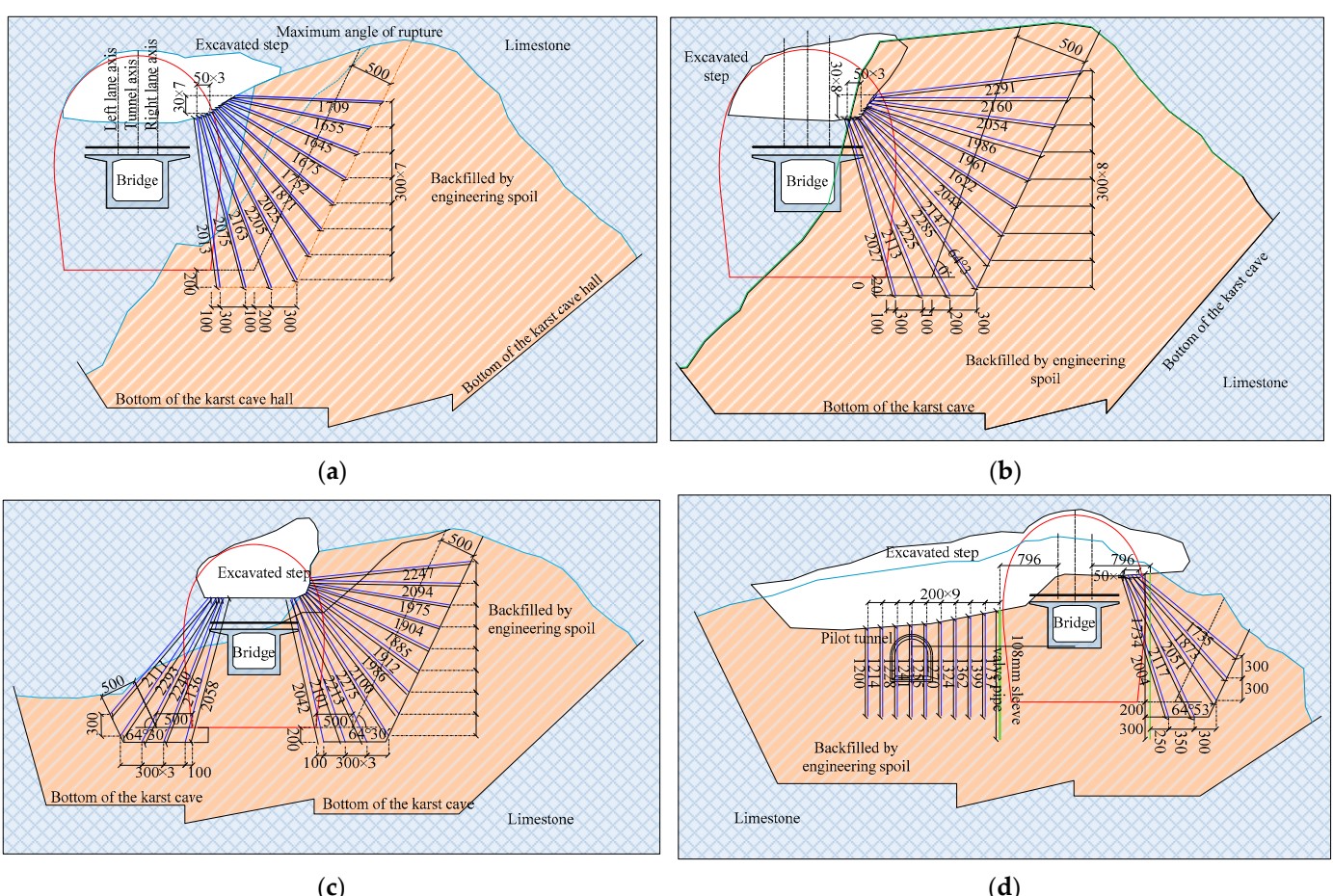

**Figure 12.** Grouting hole drilling design in typical section of (**a**) D3K279 + 883 (**b**) D3K279 + 888 (**c**) D3K279 + 903 and (**d**) D3K279 + 938.

In the processing of penetration grouting construction, the pumping pressure and slurry flow rate are the criteria for stopping penetration grouting. If the pumping pressure reaches the designed stopping pressure and remains for 900 s, the grouting can be finished. However, if the flow rate of slurry is 1.5 times of the designed grouting flow rate, and the pumping pressure is still less than the designed pressure, the gelatinization time of slurry should shorten to bring the grouting pressure up to the designed pumping pressure.

## 5. Recognition and Validation of Inadequate Grouting Section
### 5.1. Analysis of Grouting Record

The field grouting processing is recorded to recognize the inadequate reinforcement section. The drill length, actual grouted holes length, grouting volume and the grouting volume per meter of the right and left lanes of the Yujingshan tunnel are listed in Tables 4 and 5, respectively.

According to the grouting quantity of C-S slurry listed in Table 4, the grouted sections of right lane can be divide into two cases. The first case is D3K279 + 872–D3K279 + 900.5 section, in this section, the strata are mainly injected by SAC slurry, indicating that the groundwater have little influence on the section. The calculation result of Equation (5) shows that the total grouting volume per meter is theoretically no less than 1.02 m$^3$/m, and except the D3K279 + 898 and D3K279 + 900.5 sections, the total grouting volume per meter of other sections is satisfied the designed value, indicating the grouting effect of

other sections are good. The other case is D3K279 + 903–D3K279 + 935.5 section, where the C-S slurry is dominant grouting material, indicating the stratum is significantly affected by groundwater. Except for the D3K279 + 903 and D3K279 + 905.5 sections, the total grouting quantity per meter of other sections meets the designated value. The analysis results show that the D3K279 + 898–D3K279 + 905.5 surrounded by dotted line in Table 4 is the transition section of groundwater, due to the lack of grouting quantity of the section, the grouting effect should be further studied. In addition, the D3K279 + 925.5 section is only grouted by C-S slurry which plays a major role in sealing the hydraulic connection. The error value between the grouting record and the designed value is 18.63%, the grouting effect of this section needs to be analyzed by other methods.

**Table 4.** Implementation of grouting in Yujingshan tunnel right lane.

| Mileage Number | Drill Length/m | Actual Grouted Hole Length/m | Grouting Volume/m$^3$ | | | Grouting Volume Per Meter | |
|---|---|---|---|---|---|---|---|
| | | | SAC | C-S | Total | SAC | Total |
| D3K279 + 872 | 175 | 175 | 240.79 | 0 | 240.79 | 1.38 | 1.38 |
| D3K279 + 875.5 | 177 | 177 | 213.03 | 0 | 213.03 | 1.20 | 1.20 |
| D3K279 + 878 | 176.5 | 176.5 | 216.45 | 0 | 216.45 | 1.23 | 1.23 |
| D3K279 + 880.5 | 176.5 | 176.5 | 230.92 | 0 | 230.92 | 1.31 | 1.31 |
| D3K279 + 883 | 209.5 | 209.5 | 229.21 | 0 | 229.21 | 1.09 | 1.09 |
| D3K279 + 885.5 | 211 | 211 | 249.47 | 0 | 249.47 | 1.18 | 1.18 |
| D3K279 + 888 | 251.7 | 217.3 | 249.21 | 0 | 249.21 | 1.15 | 1.15 |
| D3K279 + 890.5 | 250.5 | 188.5 | 202.63 | 0 | 202.63 | 1.07 | 1.07 |
| D3K279 + 893 | 227.5 | 205 | 229.08 | 0 | 229.08 | 1.12 | 1.12 |
| D3K279 + 895.5 | 229 | 198 | 316.32 | 0 | 316.32 | 1.60 | 1.60 |
| D3K279 + 898 | 223 | 211 | 198.29 | 0 | 198.29 | 0.94 | 0.94 |
| D3K279 + 900.5 | 225.5 | 210.5 | 191.84 | 0 | 191.84 | 0.91 | 0.91 |
| D3K279 + 903 | 247 | 231 | 141.71 | 76.50 | 218.21 | 0.61 | 0.94 |
| D3K279 + 905.5 | 246.5 | 245.5 | 208.03 | 23.10 | 231.13 | 0.85 | 0.94 |
| D3K279 + 908 | 249.5 | 247.5 | 179.21 | 146.32 | 325.53 | 0.72 | 1.32 |
| D3K279 + 910.5 | 265 | 265 | 259.68 | 12.58 | 272.26 | 0.98 | 1.03 |
| D3K279 + 913 | 268 | 256 | 200.45 | 60.55 | 261.00 | 0.78 | 1.02 |
| D3K279 + 915.5 | 269.5 | 247 | 200.66 | 70.76 | 271.42 | 0.81 | 1.10 |
| D3K279 + 918 | 160.5 | 160.5 | 0 | 189.92 | 189.92 | 0 | 1.18 |
| D3K279 + 920.5 | 158 | 158 | 57.89 | 115.37 | 173.26 | 0.37 | 1.10 |
| D3K279 + 923 | 160.5 | 144.5 | 81.84 | 84.36 | 166.20 | 0.57 | 1.15 |
| D3K279 + 925.5 | 160.5 | 160.5 | 0 | 194.29 | 194.29 | 0 | 1.21 |
| D3K279 + 928 | 139 | 139 | 80.26 | 75.93 | 156.19 | 0.58 | 1.12 |
| D3K279 + 930.5 | 152 | 152 | 94.21 | 65.18 | 159.39 | 0.62 | 1.05 |
| D3K279 + 933 | 135 | 135 | 66.45 | 68.80 | 135.25 | 0.49 | 1.00 |
| D3K279 + 935.5 | 135 | 135 | 11.84 | 140.31 | 152.15 | 0.09 | 1.13 |

**Table 5.** Implementation of grouting in Yujingshan tunnel left lane.

| Grouting Section | Drill Length/m | Actual Grouted Hole Length/m | Grouting Quantity/m$^3$ | | | Grouting Quantity Per Meter/m$^2$ | |
|---|---|---|---|---|---|---|---|
| | | | SAC | C-S | Total | SAC | Total |
| D3K279 + 903 | 131 | 131 | 85.53 | 48.73 | 134.26 | 0.65 | 1.02 |
| D3K279 + 905.5 | 109 | 100.5 | 87.76 | 12.53 | 100.29 | 0.87 | 1.00 |
| D3K279 + 908 | 130.5 | 124 | 92.89 | 31.11 | 124.00 | 0.75 | 1.00 |

**Table 5.** *Cont.*

| Grouting Section | Drill Length/m | Actual Grouted Hole Length/m | Grouting Quantity/m³ | | | Grouting Quantity Per Meter/m² | |
|---|---|---|---|---|---|---|---|
| | | | SAC | C-S | Total | SAC | Total |
| D3K279 + 910.5 | 131.5 | 129.5 | 78.03 | 56.84 | 134.87 | 0.60 | 1.04 |
| D3K279 + 913 | 316.8 | 306.8 | 204.61 | 296.90 | 501.51 | 0.67 | 1.63 |
| D3K279 + 915.5 | 94 | 83 | 95.39 | 49.85 | 145.24 | 1.15 | 1.75 |
| D3K279 + 918 | 88.8 | 102 | 167.11 | 60.05 | 227.16 | 1.64 | 2.23 |
| D3K279 + 920.5 | 87 | 76 | 111.18 | 49.11 | 160.29 | 1.46 | 2.11 |
| D3K279 + 923 | 87 | 75 | 142.11 | 50.10 | 192.21 | 1.89 | 2.56 |
| D3K279 + 925.5 | 134.8 | 73.3 | 105.26 | 67.5 | 172.76 | 1.44 | 2.36 |
| D3K279 + 928 | 132.7 | 82.6 | 78.62 | 144.12 | 222.74 | 0.95 | 2.70 |
| D3K279 + 930.5 | 90.7 | 79.6 | 157.89 | 23.43 | 181.32 | 1.98 | 2.28 |
| D3K279 + 933 | 76.7 | 81.1 | 186.18 | 51.21 | 237.39 | 2.30 | 2.93 |
| D3K279 + 935.5 | 137 | 82 | 121.38 | 49.03 | 170.41 | 1.48 | 2.08 |
| D3K279 + 938 | 134.2 | 78.5 | 216.84 | 65.08 | 283.92 | 2.76 | 3.62 |
| D3K279 + 940.5 | 153.3 | 65 | 108.55 | 29.42 | 137.97 | 1.67 | 2.12 |
| D3K279 + 943 | 140 | 70 | 160.00 | 44.24 | 204.24 | 2.29 | 2.92 |
| D3K279 + 945.5 | 126 | 40 | 38.16 | 67.76 | 105.92 | 0.95 | 2.65 |
| D3K279 + 948 | 127 | 44 | 63.55 | 21.79 | 85.34 | 1.44 | 1.94 |
| D3K279 + 950.5 | 135.5 | 39 | 36.84 | 40.69 | 77.53 | 0.94 | 1.99 |
| D3K279 + 953 | 135.5 | 54.5 | 85.53 | 0 | 85.53 | 1.57 | 1.57 |
| D3K279 + 955.5 | 391 | 136.5 | 198.30 | 90.63 | 288.93 | 1.45 | 2.12 |

Table 5 is the grouting record of left lane of Yujingshan tunnel. The grouting record of left lane shows that the section are grouted by C-S and SAC slurry. However, the grouting quantity ratio of SAC slurry is larger than the C-S slurry, which indicates that the influence of groundwater on left lane is characterized by a small degree of impact, but a large range. It is noted that the total grouting quantity per meter of D3K279 + 938 section reaches 3.62 m³/m, which suggests the existence of the through crack and resulting the leakage of grout, the reason should be discussed.

*5.2. Single Hole Grouting Quantity Analysis*

According to the above analysis of grouting record, the right lane of D3K279 + 898, D3K279 + 905.5, D3K279 + 925.5 and left lane of D3K279 + 938 are selected to make further study. Figure 13 shows the grouting volume of single hole, 11 grouting holes are designed for the grouting engineering. As shown in Figure 13a, in the D3K279 + 898 section, the grouting volume of No. 4 grouting hole (28.95 m³ SAC slurry) is much more than that of No. 1–No. 3 grouting holes, and the grouting volume of No.8 is also large than that of No. 9–No. 11 grouting hole. It is indicated that the grouting hole at the side of the section is not grouted enough. The end pressure of the No. 1 grouting hole is 2.1 MPa, which is larger than the pressure at the end of another grouting hole. It is indicated that the grouting hole on the side of this section cannot produce an effective slurry stop curtain to seal the reinforcement region, and the erosion of slurry is serious. The grouting effect should be estimated by a direct method such as coring observation.

Figure 13b shows the single grouting volume of the right lane of D3K279 + 905.5 section. The grouting volume of No. 1 and No. 10 are 40.87 m³ and 22.37 m³, respectively, are larger than the adjacent grouting hole. Generally, the grouting volume of the side grouting hole is larger than that of the central grouting hole, but the end pressure is contrary to the conclusion of the grouting volume. The end pressures of No. 6 and No. 7 are 1.7 MPa and 1.8 MPa, respectively, which are larger than the other grouting holes. However, the

grouting amount are only 13.16 m$^3$ and 15 m$^3$, because the grouting hole at the side of the section is fully grouted, the lower grouting volume can produce larger pressure.

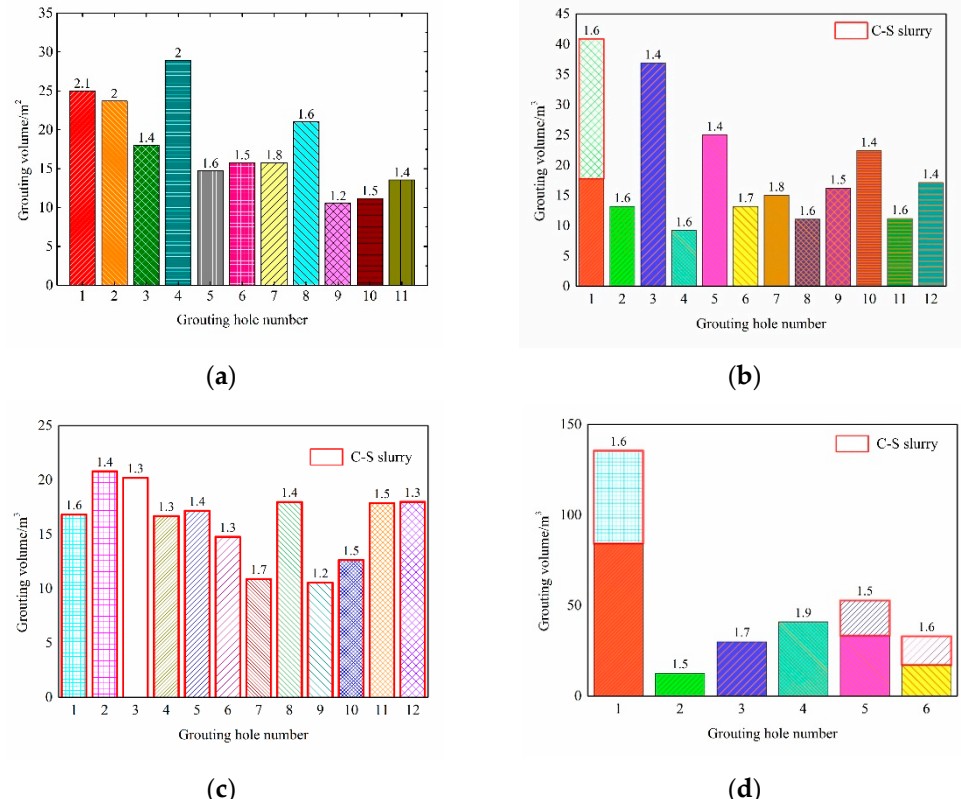

**Figure 13.** Grouting volume of single hole in (**a**) D3K279 + 898 (**b**) D3K279 + 905.5 (**c**) D3K279 + 925.5 and (**d**) D3K279 + 938 section.

Figure 13c shows the grouting record of single hole grouting in the right lane of D3K279 + 925.5 section. The end pressures of No. 1 and No. 7 are 1.6 MPa and 1.7 MPa, respectively, and the end pressures of No. 4 and No. 6 hole are 1.3 MPa, are even smaller than No. 1 hole with a value of 1.6 MPa. The grouting quantity of No. 8 hole is 17.97 m$^3$ and is almost equal to the grouting quantity of No. 11 and No. 12 holes whose grouting quantity are 17.86 m$^3$ and 17.98 m$^3$, respectively. Compared with the grouting quantity of No. 1 hole, the grouting quantity of No. 6 and No. 10 is not significantly reduction. The analysis result shows that the central hole has larger end pressure and smaller grouting quantity than the side hole; the reason for the abnormal grouting volume in this section should be investigated by direct methods.

Figure 13d shows the grouting record of D3K279 + 938 section left lane, the grouting volume of No. 1 grouting hole is larger than that of the other grouting holes with end pressure of 1.6 MPa, indicating that there is a through crack on the left side of the reinforcement region. Moreover, the data grouting amount of No. 2-No. 5 is close to the design value, which illustrates the through crack is effectively sealed by penetration grouting.

### 5.3. Validation of Recognition Result of Inadequate Reinforcement Section

For validating the recognition results of inadequate grouting section, the coring observation method is adopted in the section D3K279 + 898 and D3K279 + 925. The integrity degree of the rock core is divided into three level: when the crack range of the rock core is more than the 30% of the length of the rock core, the integrity degree of is completely fractured, when the crack range of the rock core is in 15% to 30% of the length of the rock core, that is generally fractured statues, and when the crack range of the rock core is less than 15% of the rock core length, the rock core is marked as intact. The mileage D3K279 + 898 with depression 3° and D3K279 + 925 with depression 8° sections are cored,

and the results are shown in Figure 14. Figure 14a–c are the coring results of D279 + 898 on the segment from 2 m to 8 m, from 8 m to 12 m, and from 12 m to 18 m, respectively. The coring is mainly composed of fractured grouting stone body, which cannot effectively improve the bearing capacity of surrounding rock, which illustrates that the D3K279 + 898 section is significant affected by the groundwater, and the penetration grouting is not enough to produce a curtain to prevent the influence of groundwater. The coring result of mileage D3K279 + 925 is shown in Figure 14d,e. It is observed that the D3K279 + 925 section is dominated by grouting stone body, indicating that the penetration grouting effectively reduces the water inflow and improves the strength of surrounding rock.

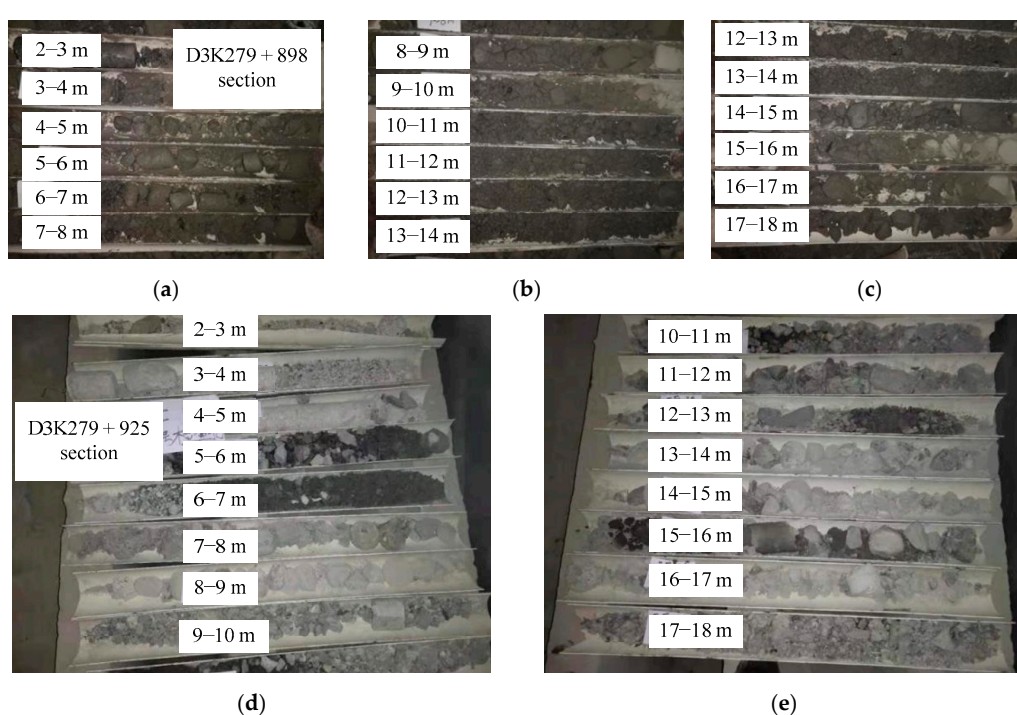

**Figure 14.** The cores of D3K279 + 898 and D3K279 + 925 section. (**a**) D3K279 + 898 section 2–8 m. (**b**) D3K279 + 898 section 8–14 m. (**c**) D3K279 + 898 section 12–18 m. (**d**) D3K279 + 925 section 2–10 m. (**e**) D3K279 + 925 section 10–18 m.

According to the recognition result of inadequate reinforcement section, it is obtained that the D3K279 + 898 section is significantly affected by groundwater, the penetration grouting at D3K279 + 898 section cannot seal the hydraulic channel, and supplementary grouting is needed. In addition, in order to reduce the influence of groundwater, the D3K279 + 898 and D3K279 + 900.5 should be firstly complement grouted by C-S slurry to further seal the hydraulic crack, and then grouted by SAC slurry to improve the bearing capacity.

*5.4. Application Effect Evaluation of Countermeasure by Field Monitor*

Through field monitor, the application effect of countermeasure of grouted engineering spoil backfill giant karst cave is analyzed. Figure 15 shows the monitoring results of the horizontal convergence of Yujingshan tunnel. Figure 15a shows the horizontal convergence on the upper step. It shows the tendency of gently increasing first, then rapidly increasing, and finally reaching a stable state. The maximum horizontal convergence is located at D3K279 + 885 section, with a value of 34.47 mm. The D3K279 + 896 section also has a large horizontal convergence, with a maximum value of 36.57 mm. In the rapid growth stage, the convergence velocity of the D3K279 + 885 and D3K279 + 896 are 0.91 mm/d and 1.25 mm/d, respectively. The results show that the maximum value and maximum velocity appear at D3K279 + 896 section, indicating that the grouting effect is not preferable. The horizontal convergences of most sections are only 8–13 mm, and the convergence

velocity is much lower than of the D3K279 + 885 and D3K279 + 896 sections, which is about 0.45 mm/d at the same stage.

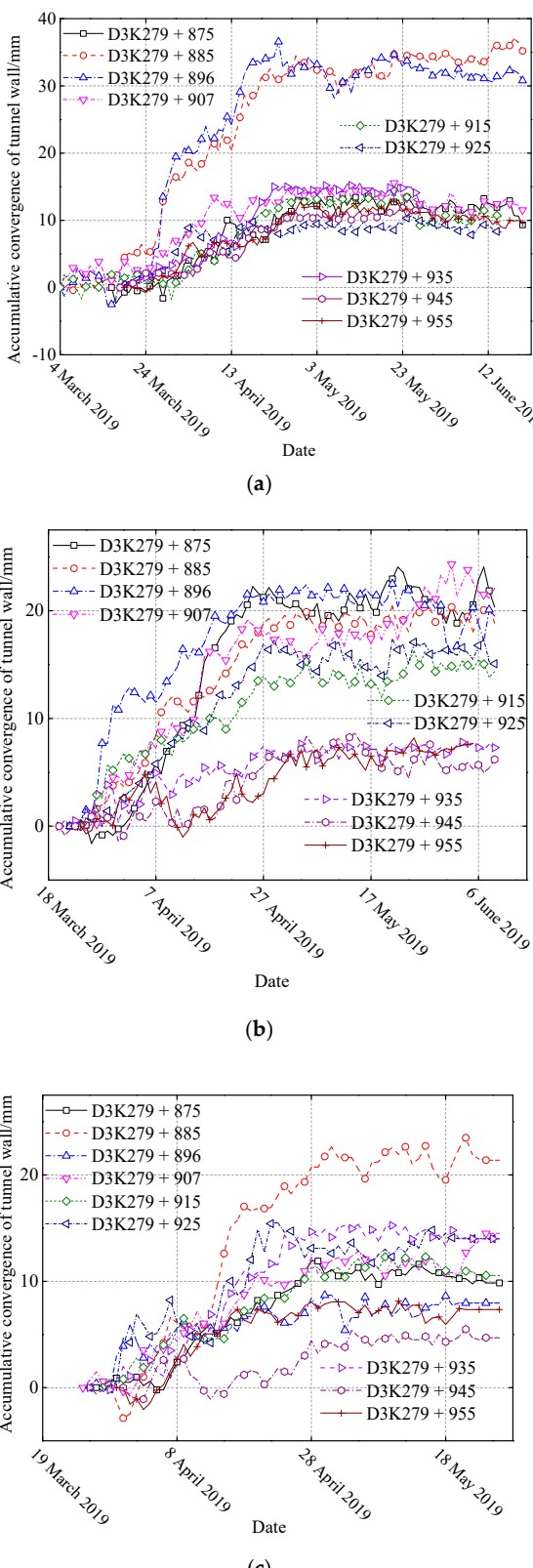

**Figure 15.** Horizontal convergence of Yujingshan tunnel on the (**a**) upper (**b**) middle and (**c**) lower steps.

Figure 15b shows the monitor results of middle step excavation. The mileage D3K279 + 875–D3K279 + 925 produces horizontal convergence with 13 mm to 26 mm. Among them, D3K279 + 896 section maintains a large convergence with a value of 22.42 mm and a convergence velocity of 0.75 mm/d. The D3K279 + 875 section produces the maximum horizontal convergence of 24.05 mm and a convergence velocity of 0.78 mm/d. The monitoring results show that the horizontal convergences of D3K279 + 875–D3K279 + 925 section (7–9 mm) are larger than those of D3K279 + 935–D3K279 + 955 section. The deformation characteristic is that it increases slowly at first, then increases rapidly, and finally gradually tends to a stable state, similar to the excavation of upper step. However, it produces a shorter slow increasing stage and a longer rapid increasing stage than the upper step, and the maximum convergence velocity is 0.78 mm/d during rapidly increasing stage. All the monitor sections produce a controlled horizontal convergence which is smaller than 30 mm. The monitor results indicate that the penetration grouting can effectively reduce the horizontal convergence to allowable value, and improve the convergence velocity to a gentle state.

As shown in Figure 15c, the maximum horizontal convergence of the lower step excavation occurs in D3K279 + 885 section with a value of 23.49 mm, and a convergence velocity is 0.78 mm/d. The horizontal convergences of other sections are less than 15 mm, and a convergence velocity of only 0.56 mm/d. The convergences of different sections show that after a short term rapid increasing stage, the horizontal convergence is in a stable state, and its value is less than the allowable value. The monitor results of lower step indicate that the lower step region is adequately reinforced by penetration grouting, its convergence value is low, and it reaches a stable state faster than the upper and middle steps. The variation law of displacement of surrounding rock show the similar tendency with previous literature [41,42], which demonstrate the great grout effect is achieved.

## 6. Conclusions

By using the interface element to connect the backfill body and karst cave, an effective FLAC$^{3D}$ finite difference model is established. The variation characteristics of the surrounding rock deformation and the mechanical response of the support structure before and after grouting are analyzed. The feasibility and rationality of using grouting to pretreat the backfill body to ensure the safety of tunnel excavation are analyzed. The inadequate grouting section was recognized by using the field grouting record analysis and the single hole grouting quantity analysis, and recognition results was verified by the coring observation method. Finally, the grouting effect is evaluated by using field monitoring results of horizontal convergence. The following conclusions are obtained.

The position relationship between tunnel and karst cave has a great influence on the displacement development law of surrounding rock and the mechanical response of the supporting structure. The karst cave located on the single and both sides of the tunnel are defined as I-type and II-type tunnel, respectively. Before grouting, the surrounding rock has a large displacement. In the backfill region of the I-type and II-type tunnels, the surrounding rock at the side wall and bottom of the tunnel has a large deformation, while the backfill body in the I-type tunnel away from the side of the tunnel produces a similar landslide phenomenon. Large bending moments are generated in the backfill region and contact region of I-type and II-type tunnels. In the I-type tunnel, the karst cave is located on one side of the tunnel, resulting in large torque in the backfill region and contact region. However, the distribution of karst cave in II-type tunnel is more symmetrical than that in I-type tunnel, the torque is only generated at the bottom of the tunnel. The deformation of the support structure is related to the excavation method. For I-type and II-type tunnels, the large deformation is occurred at the foot of each step and the bottom of the tunnel, while the deformation at the vault is almost 0. Grouting can reduce the displacement of the surrounding rock in I-type and II-type by 70–90%, the range of stress redistribution in the surrounding rock is decreased, and the displacement in the landslide region of I-type is reduced by 43%. The bending moment of the support structure within the backfill body

is reduced by 80–90%, the torque is reduced to negligible, and the large deformation of the support structure at the foot of each step and the bottom of the tunnel is effectively controlled. Excepted for D3K279 + 898 section, other sections are sufficient grouted, with small horizontal convergence which is less than 30 mm and short stabilization time which is less than 20 days, good grouting effect is achieved. D3K279 + 898 section is the transition interval affected by groundwater. The grouting material on the section is cement-based single slurry, and the grouting quantity is 0.92 $m^3$/m, which is less than the designed value. The single hole grouting quantity is characterized by large grouting quantity in the middle hole and large end pressure of surrounding hole. Core observation reveals that the rock core is broken and has large moisture content on D3K279 + 898, D3K279 + 898 is an inadequate grouting section. The hydraulic connection should be sealed with cement-sodium silicate slurry and then reinforced with the cement slurry.

**Author Contributions:** Conceptualization, P.P. and F.P.; methodology, Z.S.; software, P.P. and Z.S.; validation, Z.S., D.Z. and F.P.; formal analysis, P.P. and Z.S.; investigation, F.P.; data curation, F.P.; writing—original draft preparation, P.P.; writing—review and editing, P.P. and Z.S.; visualization, Z.S.; supervision, D.Z.; project administration, D.Z.; funding acquisition, D.Z. All authors have read and agreed to the published version of the manuscript.

**Funding:** This research was funded by National Natural Science of China, grant number 51738002.

**Institutional Review Board Statement:** Not applicable.

**Informed Consent Statement:** Not applicable.

**Data Availability Statement:** Not applicable.

**Conflicts of Interest:** The authors declare no conflict of interest. The funders had no role in the design of the study; in the collection, analyses, or interpretation of data; in the writing of the manuscript; or in the decision to publish the results.

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
