# Peer review of "Grouting for Tunnel Stability Control and Inadequate Grouting Section Recognition: A Case Study of Countermeasure of Giant Karst Cave"

_applsci, doi:10.3390/app122311895_

Round 1

Reviewer 1 Report

In my opinion, the word "diffusion" - verses 389, 391, 392 - is unfortunate and should come down to others. The described process is not classical diffusion (Fick's laws of diffusion).

Reviewer 2 Report

General comments

The general characterization of the study area is absent. The available information “The tunnel passes through a geological section of mudstone clamp sandstone. The surrounding rock grade is IV.” it is not enough. How can we discuss the results? Do authors consider the material as homogeneous?

None of the data discussed in sections 3.2, 3.3, 3.4 and 5 is somewhat compared with similar cases. Just the description of the results is not enough. Study cases in limestones should be considered during the discussion of the results.

Specific comments

Line 129 - “the location of the Yujingshan tunnel developed Niuer slope No.1 fault, Niuer slope No.2 fault and Weixin fault”

These faults are missing in the Fig.

Line 135 - “According to the engineering construction scheme, the karst cave will be backfilled with engineering spoil”

Please, add more information about the backfill.

Line 144 - “There are 5 proved inlet total flux is 1.2 m3/s~5.0 m3/s, and 9 outlet, the maximum observed flux is about 21.16 m3/s.”

Please, add the source of such information.

Line 168 - “The size of tunnel infrastructure is 17.22 168 m× 25.21 m.”

Such information should be presented in an earlier chapter.

Line 171 - “According to the geological survey report, the computational region can be divide into two layers, the first layer is dominated by the fragment stone, and the second layer is composed of limestone. The physical and mechanics parameters of stratum are listed in Table.1”

I suggest moving to chapter 2.

Line 173 - “The physical and mechanics parameters of stratum are listed in Table.1.”

Please, add the source of such information.

Line 211 - “surrounding rock in the upper step region is softening by the groundwater.”

The full characterization of the cave and limestone is missing.

Line 402 - “the result of field test shows that when the concentration of slurry is 0.5, grouting time is 900 s and the pumping pressure is 2 MPa”

Source?

Line 463 - “The analysis results shows that the D3K279+898 – D3K279+905.5 is the transition section of groundwater”

Can you add colour to the tables? The transitions will be identified more easily.

Line 521 - “the water volume fraction of D3K279+898 section is extremely high, it illustrates that the D3K279+898 section is significant affected by the groundwater”

We don’t have such information.

Line 525 - “is shown in Figs. 10d and e”

Fig 13? Check it.

Figures

Figure 11 - A legend will improve the readability of the figure.

Figure 12 - The lettering is too small.

Reviewer 3 Report

There are lots of English language problems in the entire manuscript. For example, in the abstract, the line 4, "In order to ensure the countermeasure...." should be "In order to ensure that the countermeasure......". 

Line 8 in the abstract, "...evaluated by field monitor of horizontal...." is wrongly presented. The word monitor should be monitoring. Authors need to conduct an extensive English editing if this has to be considered for publication. 

The coring observation method should be extensively discussed in the method to enable research replication. 

Meanwhile the results should be compared with literature findings. 

Reviewer 4 Report

The current paper studied tunneling in karst zone by 3d numerical modelling. Even though the subject of the paper is very interesting for geotechnical engineers and practical applications, however, the it suffers a lack of comprehensive investigation and scientific conceptualization. The scientific structure of the manuscript should be improved. The manuscript seems a report on tests, rather than a paper. Besides, the paper needs to be revised by a native speaker as some of the sentences are confusing and some words seem not to be correctly applied.

The current paper studied tunneling in karst zone by 3d numerical modelling. Even though the subject of the paper is very interesting for geotechnical engineers and practical applications, however, the it suffers a lack of comprehensive investigation and scientific conceptualization. The scientific structure of the manuscript should be improved. The manuscript seems a report on tests, rather than a paper. Besides, the paper needs to be revised by a native speaker as some of the sentences are confusing and some words seem not to be correctly applied.

Round 2

Reviewer 2 Report

General comments

Just the description of the results is not enough. Study cases in limestones should be considered during the discussion of the results.

Reviewer 3 Report

The revisions are sufficient 

Reviewer 4 Report

The manuscript seems a report on tests, rather than a paper and van be suitable for publication in  a journal

Round 3

Reviewer 2 Report

After text editing the manuscript could be accepted.